# Nitration of 2,6,8,12-Tetraacetyl-2,4,6,8,10,12-Hexaazaisowurtzitane Derivatives

**DOI:** 10.3390/ma15227880

**Published:** 2022-11-08

**Authors:** Maya V. Chikina, Daria A. Kulagina, Sergey V. Sysolyatin

**Affiliations:** Laboratory for Medicinal Chemistry, Institute for Problems of Chemical and Energetic Technologies, Siberian Branch of the Russian Academy of Sciences (IPCET SB RAS), 659322 Biysk, Russia

**Keywords:** nitrolysis, CL-20, 2,6,8,12-tetraacetyl-2,4,6,8,10,12-hexaazaisowurtzitane, mixed sulfric/nitric acids, nitric acid

## Abstract

The nitration of novel bioactive derivatives of 2,6,8,12-tetraacetyl-2,4,6,8,10,12-hexaazaisowurtzitane in different nitrating systems was examined. The yield of CL-20, the known product from the nitration of hexaazaisowurtzitane compounds, was found to depend on the nature of substituents at the 4,1 positions and on the composition of the nitrating mixture.

## 1. Introduction

2,4,6,8,10,12-Hexaazaisowurtzitane derivatives are cage polycyclic amines with a unique structure that makes them appealing for study in different fields of organic chemistry. These compounds are traditionally produced by condensation of glyoxal with benzylamine [1] followed by the modification to the target structures. Indeed, a method is known for the synthesis of hexaazaisowurtzitane compounds by transamination of N,N′-di-tert-butyl-1,2-ethanediimine with amines [2].

These compounds have become best known as intermediate products for the synthesis of the powerful explosive CL-20 (2,4,6,8,10,12-hexanitro-2,4,6,8,10,12-hexaazaisowurtzitane, 1) [3,4,5,6,7,8,9,10]. However, a range of 2,6,8,12-tetraacetyl-2,4,6,8,10,12-hexaazaisowurtzitane derivatives has presently been created that exhibit significant biological activity combined with low toxicity [11,12,13,14,15,16,17,18,19,20], making them promising for a wide range of use.

Here, we explored the nitration process of innovative compounds that exhibit an analgesic activity, these are:-4,10-di(2-ethoxyacetyl)-2,6,8,12-tetraacetyl-2,4,6,8,10,12-hexaazaisowurtzitane (**2**);-4-(3,4-dibromothiophenecarbonyl)-2,6,8,12-tetraacetyl-2,4,6,8,10,12-hexaazaisowurtzitane (thiowurtzine, **3**);-4-(3,4-dibromothiophenecarbonyl)-10-(2-ethoxyacetyl)-2,6,8,12-tetraacetyl-2,4,6,8,10,12-hexaazaisowurtzitane (**4**);-4-(3,4-dibromothiophenecarbonyl)-2,6,8,10,12-pentaacetyl-2,4,6,8,10,12-hexaazaisowurtzitane (**5**);-4,10-bis((±)-5-benzoyl-2,3-dihydro-1Н-pyrrolo [1,2-а]pyrrol-1-carbonyl)-2,6,8,12-tetraacetyl-2,4,6,8,10,12-hexaazaisowurtzitane (**6**).

## 2. Materials and Methods

^1^H and ^13^C NMR spectra were recorded on a Bruker Avance III spectrometer (Bruker Corporation, Billerica, MA, USA). ^1^H NMR spectra were acquired at 400.13 MHz, while ^13^C NMR spectra were taken at 100.61 MHz. The measurements were conducted at 298 K, unless otherwise stated. The spectra were calibrated using residual solvent signals (DMSO-*d*_6_: 2.50 ppm for ^1^H, 39.5 ppm for ^13^C). All NMR spectra of the new compounds are shown in the Appendix A. IR spectra (KBr): Simex FT-801 FTIR spectrometer (Simex, Novosibirsk, Russia). Elemental analyses were done on a Thermo Scientific Flash EA1112 CHNS elemental analyzer (Thermo Fisher Scientific, Waltham, MA, USA) for carbon, hydrogen, nitrogen, and oxygen contents. The reagents were procured from commercial sources and used as received unless otherwise stated. The commercially available compounds were used without additional purification unless otherwise stated. 


**A general procedure for the nitration of 2,6,8,12-tetraacetyl-2,4,6,8,10,12-hexaazaisowurtzitane derivatives (2–6) in mixed H2SO4-HNO3 (30/70) at a modulus of 30 was as follows:**


In a 100-mL three-neck flask equipped with an oil bath, a thermometer, and a reflux condenser, nitric acid (28 mL, 42 g) was poured into sulfuric acid (11.5 mL, 18 g) with stirring at a temperature not above 30 °C, and afterwards 2–6 (2 g) were added portionwise at 20–25 °C and held for 30 min at this temperature until completely dissolved. The reaction mixture was then heated to 70–73 °C and held for 6 h. Once the holding was completed, the reaction mixture was cooled to 40 °C and poured into ice (300 g); the precipitate was collected by filtration and air-dried. The resultant product was analyzed by HPLC. 


**A general procedure for the nitration of 2,6,8,12-tetraacetyl-2,4,6,8,10,12-hexaazaisowurtzitane derivatives (2–6) in mixed NH4NO3-HNO3 (25/75) at a modulus of 10 was as follows:**


In a 100-mL three-neck flask equipped with an oil bath, a thermometer, and a reflux condenser, nitric acid (20 mL, 30 g) was added into ammonium nitrate (10 g) with stirring at a temperature of no more than 25 °C; afterward, 2–6 (4 g) were added portion-wise and held for 30 min at this temperature until fully dissolved. The reaction mixture was then heated to 115–120 °C and held for 6 h. After the holding was completed, the reaction mixture was cooled to 40 °C and poured into ice (150 g); the precipitate was collected by filtration and air-dried. The resultant product was analyzed by HPLC. 

**4-(2-Nitro-3,4-dibromothiophenecarbonyl)-2,6,8,10,12-pentanitro-hexaazaisowurtzitane 7**: ^1^H NMR (DMSO-*d*_6_, *δ*, ppm) 6.98–7.23 (m, 2CH, CH-Iw), 7.92–8.09 (m, 4CH, CH-Iw). ^13^C NMR (DMSO-*d*_6_, *δ*, ppm) 71.16, 71.57, 74.09 (CH, Iw), 117.91 (C, CBr), 118.71 (C, CBr), 132.64 (C, CS), 148.97 (C, CNO_2_), 160.28 (CO). FTIR (KBr): 593, 710, 757, 784, 824, 867, 897, 966, 1048, 1097, 1180, 1283, 1402, 1529, 1617, 1702, 3032, 3061 cm^−1^. Elementary analysis: calcd C 18.71, H 0.86, N 23.80, S 4.54; found C 18.96, H 0.83, N 23.71, S 4.61 (see Appendix A).

**4-(2-Nitro-3,4-dibromothiophenecarbonyl)-10-acetyl-2,6,8,12-tetranitro-2,4,6,8,10,12-hexaazaisowurtzitane 8**: ^1^H NMR (DMSO-*d*_6_, *δ*, ppm) 2.44 (s, 3CH, CH_3_), 6.96–7.24 (m, 3CH, CH-Iw), 7.71–8.09 (m, 3CH, CH-Iw). ^13^C NMR (DMSO-*d*_6_, *δ*, ppm) 20.85 (CH_3_), 66.62, 70.33, 71.15, 71.57, 73.69, 74.08 (CH, Iw), 117.83 (C, CBr), 118.32 (C, CBr), 133.18 (C, CS), 148.95 (C, CNO_2_), 160.96 (CO), 169.80 (CO, CH_3_CO). FTIR (KBr): 603, 661, 751, 770, 825, 889, 964, 1103, 1174, 1327, 1400, 1489, 1585, 1691, 3033 cm^−1^. Elementary analysis: calcd C 22.21, H 1.29, N 21.91, O 27.30, S 4.56; found C 22.03, H 1.50, N 21.15, S 4.38 (see Appendix A).

**2,4,6,8,10,12-Hexanitro-2,4,6,8,10,12-hexaazaisowurtzitane 1**: ^1^H NMR (DMSO-*d*_6_, *δ*, ppm) 7.99 (s, 4H, CH-Iw), 8.09 (s, 2H, CH-Iw). ^13^C NMR (DMSO-*d*_6_, *δ*, ppm) 71.56, 74.13 (CH, Iw).

## 3. Results and Discussion

Based on the known data on optimum conditions for the synthesis of 1 [9], we chose the following nitrating systems for the study: mixed sulfuric/nitric acids (30:70), a classical option, and nitric acid with added 25% ammonium nitrate providing milder reaction conditions.

Nitrolysis of 2 in mixed sulfuric/nitric acids at 70 °C was quite fast and furnished 1, with the simultaneous destruction of the final product. In case nitric acid with ammonium nitrate was employed at 110–120 °C, the yield of product 1 was 67% (Figure 1).

When **3** was nitrated with nitric acid/ammonium nitrate at 110–120 °C for 8 h, the reaction products were **1** and 4-(2-nitro-3,4-dibromothiophenecarbonyl)-2,6,8,10,12-pentanitrohexaazaisowurtzitane (**7**) (Figure 2) in 22% and 41% yields, respectively. 

An increase in the reaction length to 16 h resulted in the higher destruction of the starting molecule of **3**, and the reaction product was **1** mixed with **7** in yields of 7.5% and 18%, respectively.

In the case of nitration with mixed sulfuric/nitric acids at 70 °C, the reaction product was **1** in a 33% yield, while compound **7** in the reaction mass was not documented.

Nitrolysis of compound 4 with nitric acid/ammonium nitrate at 110–120 °C also gave mixed compounds **1** and **7** in 34% and 25% yields, respectively (Figure 2). Mixed sulfuric/nitric acids used for the nitration of 4 led to a fast-occurring process with simultaneous destruction of the final product.

By nitration of compound 5 with nitric acid/ammonium nitrate at 110–120 °C for 6 h, we isolated 4-(2-nitro-3,4-dibromothiophenecarbonyl)-10-acetyl-2,6,8,12-tetranitro-2,4,6,8,10,12-hexaazaisowurtzitane (**8**) in a 61% yield (Figure 3). When the holding time was raised, no formation of product 1 was documented. In case mixed sulfuric/nitric acids were utilized, product 8 was not observed to be formed, with the final reaction product being **1** in a 34% yield.

Nitrolysis of compound **6** with mixed nitric acid/ammonium nitrate at 110–120 °C furnished compound **1** in a 30% yield (Figure 4). The use of mixed sulfuric/nitric acids to nitrate 6 resulted in a fast-occurring process with simultaneous destruction of the final product.

## 4. Conclusions

Thus, the present study demonstrated that the nitration of bioactive derivatives of 2,6,8,12-tetraacetyl-2,4,6,8,10,12-hexaazaisowurtzitane with mixed sulfuric/nitric acids in a ratio of 30:70 was fast and led basically to the degradation of the starting and final products. The process carried out under milder conditions (nitric acid mixed with 25% ammonium nitrate) afforded CL-20 as the product of complete nitration in a yield ranging from 30 to 67%.

Indeed, it was discovered that the compound with linear substituents at the 4,10 positions underwent nitration most easily. Nitration of the compounds bearing the thiophene moiety at one position proceeded through an intermediate stage to generate derivatives comprising the mononitro thiophene moiety. 

It is worth noting individually that the yield of CL-20 as the end product is not technologically significant and hence the wide application of the studied promising hexaazaisowurtzitane compounds with an analgesic activity will make it impossible to use the same for terrorism purposes.

## Data Availability

Not applicable.

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
