# Peer review of "Nitration of 2,6,8,12-Tetraacetyl-2,4,6,8,10,12-Hexaazaisowurtzitane Derivatives"

_materials, 2022, doi:10.3390/ma15227880_

Round 1

Reviewer 1 Report

In the present communication, the authors explored the process of nitration of certain bioactive derivatives of 2,6,8,12-tetraacetyl-2,4,6,8,10,12-hexaazaisowurtzitane. Five derivatives with analgesic activity were examined. In general, the results and discussion are informative and the manuscript is organized and well written. Thus, this paper is suitable for publishing in Materials. However, there are a few minor comments that should be considered.

- Some typos need to be eliminated such as: line 29 algesic activity ---> analgesic activity.

- I suggest removing the word "new" from the title since the molecules are not new, only the nitration procedure is original.

- Supporting Materials file is not provided.

Author Response

The response to Reviewer 1 has been uploaded as a Word file.

Reviewer 2 Report

I have the following comments about this manuscript:

- the supporting information cannot be downloaded so the purity and the characterisation of the compounds cannot be assessed

- I received a review request for a manuscript to be included in the "Special Issue: Fabrication, Characterization and Application of High-Energy Material" in Materials so I am not sure how the present manuscript fits this SI. Especially when in the intro they are discussed for the analgesic activity

- should the authors include a warning about the danger of nitration of this class of explosive? Precautions should be included with this synthesis, especially at these scales

- my ethical concerns are directed towards both analgesic activity as it seems not a great idea to give this compounds to patients and also to develop poly nitro explosive compounds

Author Response

The response to Reviewer 2 has been uploaded as a Word file.
